# Operation of spinal sensorimotor circuits controlling phase durations during tied-belt and split-belt locomotion after a lateral thoracic hemisection

**Ilya A Rybak[1]\*, Natalia A Shevtsova[1], Johannie Audet[2], Sirine Yassine[2], Sergey N Markin[1], Boris I Prilutsky[3], Alain Frigon[2]\***

[1]Department of Neurobiology and Anatomy, College of Medicine, Drexel University, Philadelphia, United States; [2]Department of Pharmacology-Physiology, Faculty of Medicine and Health Sciences, Centre de Recherche du CHUS, Université de Sherbrooke, Sherbrooke, Canada; [3]School of Biological Sciences, Georgia Institute of Technology, Atlanta, United States

**\*For correspondence:**
iar22@drexel.edu (IAR);
Alain.Frigon@USherbrooke.ca
(AF)

**Competing interest:** The authors declare that no competing interests exist.

## eLife Assessment

This **important** modeling study alters a previous model of the intact cat spinal locomotor network to simulate a lateral hemi-section of the spinal cord. The modeling and experimental work described provide **convincing** evidence that this model is capable of qualitatively predicting alterations to the swing and stance phase durations during locomotion at different speeds on intact or split-belt treadmills. This paperarticle will interest neuroscientists studying vertebrate motor systems, including researchers working on motor dysfunction after spinal cord injury.

**Abstract** Locomotion is controlled by spinal circuits that interact with supraspinal drives and sensory feedback from the limbs. These sensorimotor interactions are disrupted following spinal cord injury. The thoracic lateral hemisection represents an experimental model of an incomplete spinal cord injury, where connections between the brain and spinal cord are abolished on one side of the cord. To investigate the effects of such an injury on the operation of the spinal locomotor network, we used our computational model of cat locomotion recently published in *eLife* (Rybak et al., 2024) to investigate and predict changes in cycle and phase durations following a thoracic lateral hemisection during treadmill locomotion in tied-belt (equal left-right speeds) and split-belt (unequal left-right speeds) conditions. In our simulations, the 'hemisection' was always applied to the right side. Based on our model, we hypothesized that following hemisection the contralesional ('intact', left) side of the spinal network is mostly controlled by supraspinal drives, whereas the ipsilesional ('hemisected', right) side is mostly controlled by somatosensory feedback. We then compared the simulated results with those obtained during experiments in adult cats before and after a mid-thoracic lateral hemisection on the right side in the same locomotor conditions. Our experimental results confirmed many effects of hemisection on cat locomotion predicted by our simulations. We show that having the ipsilesional hindlimb step on the slow belt, but not the fast belt, during split-belt locomotion substantially reduces the effects of lateral hemisection. The model provides explanations for changes in temporal characteristics of hindlimb locomotion following hemisection based on altered interactions between spinal circuits, supraspinal drives, and somatosensory feedback.

## Introduction

The basic control of flexor and extensor activities during locomotion is performed by circuits located in the spinal cord, including those of the central pattern generator, that receive inputs from the brain and sensory feedback from peripheral receptors (*Grillner, 1981*; *Rossignol et al., 2006*; *Kiehn, 2006*; *Frigon, 2017*; *Frigon et al., 2021*). It has been hypothesized that somatosensory feedback plays a major role in the control of locomotion at lower speeds, and that this role decreases as speed increases (*Full and Koditschek, 1999*; *Holmes et al., 2006*; *Ijspeert and Daley, 2023*). Consequently, different forms of spinal cord injuries (SCIs) may affect spinal circuits and/or their speed-dependent control by supraspinal and somatosensory mechanisms, hence changing the locomotor pattern and related flexor and extensor activities. In humans, most SCIs are not complete and some pathways that communicate between the brain and spinal cord remain intact (*Harkema et al., 2011*; *Angeli et al., 2014*). A frequently used experimental model of incomplete SCI in animal models is a lateral hemisection, performed by surgically sectioning half the spinal cord in the transverse plane. A lateral hemisection, which reproduces features of the Brown-Séquard syndrome observed in humans, abolishes all spinal pathways on one side of the cord (*Shams et al., 2024*). This effectively cuts off bidirectional communication between the brain and spinal cord on one side. Several studies have shown how the locomotor pattern adjusts following a lateral hemisection at thoracic levels, mainly in rats and cats (*Kato et al., 1984*; *Helgren and Goldberger, 1993*; *Webb and Muir, 2002*; *Barrière et al., 2008*; *Frigon et al., 2009*; *Barrière et al., 2010*; *Filli et al., 2011*; *Doperalski et al., 2011*; *Martinez et al., 2011*; *Martinez et al., 2012*; *Martinez et al., 2013*; *Thibaudier et al., 2017*; *Lecomte et al., 2022*; *Audet et al., 2023*). The main feature is that the locomotor pattern becomes asymmetric with different stance and swing phase durations for the contralesional and ipsilesional hindlimbs. However, changes in sensorimotor interactions following a lateral hemisection and other incomplete SCIs, as well as their impact on the control of phase durations/transitions, remain poorly understood.

Using experimental data obtained during treadmill locomotion from intact cats and cats with a complete thoracic SCI (i.e. ,spinal transection), we previously built a computational model of the neural control of cat hindlimb locomotion (*Rybak et al., 2024*). We used this model to investigate how supraspinal inputs and somatosensory feedback interact with spinal circuits to control phase durations over a range of speeds during tied-belt (equal left-right speeds) and split-belt (unequal left-right speeds) treadmill locomotion. Based on that study, we proposed that spinal circuits can operate in different regimes depending on the speed of locomotion and available inputs from supraspinal structures and somatosensory feedback. Specifically, in the intact condition, spinal circuits operated in a *state-machine regime* (an external signal is necessary for the extension-to-flexion transitions) at slow speeds before transitioning to a *flexor-driven regime* (locomotor activity is defined by the intrinsic oscillations of the flexor half-center) and then to a *classical half-center regime* (the oscillations require mutual inhibition of both half-centers) at higher speeds. The relative contributions from supraspinal drive and somatosensory feedback to the control of flexor and extensor phase durations changed depending on speed and the spinal network's operating regime. In spinal-transected cats, however, spinal circuits operated exclusively in a state-machine regime, requiring sensory feedback from the limbs to regulate flexor and extensor phase durations across speeds.

In the present study, we used the above computational model to simulate the effects of a thoracic lateral hemisection and predict speed-dependent changes in cycle and phase durations during tied-belt and split-belt locomotion after hemisection. We hypothesized that following hemisection the contralesional ('intact') side is mostly controlled by supraspinal drives, whereas the ipsilesional ('hemisected') side is mostly controlled by somatosensory feedback. We then collected data in adult cats before and after a mid-thoracic lateral hemisection in the same locomotor conditions and compared experimental results with modeling predictions. The simulated and experimental results concerning the cycle, swing, and stance durations during tied-belt and split-belt treadmill locomotion across speeds were qualitatively similar. The present findings extend the current understanding of the operation of the spinal locomotor network and the specific roles of supraspinal and somatosensory inputs in the control of this network before and after SCI.

# Results

## Modeling the effects of thoracic lateral hemisection on cycle and phase durations during tied-belt and split-belt locomotion

### Model description

In the present study, we used our computational model of the neural control of locomotion described in detail in our *eLife* paper (*Rybak et al., 2024*). The schematic of the intact model with the spinal cord circuits, supraspinal drives, and somatosensory feedback is shown in *Figure 1A*. The model describes neuronal circuits involved in the control of, and interactions between, the cat hindlimbs during treadmill locomotion. Briefly, the spinal circuitry in the model includes two rhythm generators (RGs), each consisting of two conditional bursters, representing flexor (F) and extensor (E) half-centers, respectively, inhibiting each other through inhibitory neurons, InF and InE. The left and right RGs interact via a series of commissural interneuronal (CIN) pathways mediated by different sets of genetically identified commissural (V0$_D$, V0$_V$, and V3) and ipsilaterally projecting (V2a) interneurons and some hypothetical inhibitory interneurons (Ini). The organization of these interactions was directly drawn from our earlier models (*Rybak et al., 2015*; *Shevtsova et al., 2015*; *Danner et al., 2017*; *Danner et al., 2019*; *Zhang et al., 2022*; *Shevtsova et al., 2022*) and was based on the results of previous molecular/genetic studies in mice or proposed to explain and reproduce different aspects of the neural control of locomotion in these studies.

As described in detail in our previous paper (*Rybak et al., 2024*), each RG can operate in one of three regimes: *state-machine*, *flexor-driven*, and *classical half-center* regimes. Specifically, the non-oscillatory state-machine regime requires an external input (e.g., feedback signal) to switch from extension to flexion. In the flexion-driven regime, the RG oscillations are defined by intrinsic bursting properties of the F half-center, whereas in the classical half-center regime both half-centers and their mutual inhibition are critically involved in RG oscillations.

The left and right RGs receive, correspondingly, left ($\alpha_L$, $\gamma_L$) and right ($\alpha_R$, $\gamma_R$) 'supraspinal' drives and left (l-SF-E1, l-SF-E2) and right (r-SF-E1, r-SF-E2) sensory feedback from the hindlimbs. The SF-E1 feedback represents sensory feedback from spindle afferents of hip flexor muscles, whose activity increases when these muscles are stretched when the hip extends during stance (*Grillner and Rossignol, 1978*; *Kriellaars et al., 1994*; *Pearson, 2004*; *Pearson, 2008*; *Klishko et al., 2021*; *Frigon et al., 2021*). This feedback promotes the extensor-flexor phase transition via excitatory input to the ipsilateral F half-center and inhibitory input to the contralateral F half-center and increases with increasing treadmill speed. The SF-E2 feedback simulates in a simplified form the involvement of force-dependent group Ib afferent excitatory feedback from limb extensor muscles to the ipsilateral E half-center, reinforcing extensor activity and weight support during stance (*Duysens and Pearson, 1980*; *Pearson and Collins, 1993*; *Gossard et al., 1994*; *Frigon et al., 2021*).

The supraspinal drives ($\alpha_L$ and $\alpha_R$) presynaptically inhibit inputs from all ipsilateral somatosensory feedback types (see *Figure 1A*), hence reducing their influence on the operation of RGs. As a result, in the intact model at relatively high drives ($\alpha_L$ and $\alpha_R \geq 0.35$), the locomotor pattern is mostly defined by supraspinal drives and only slightly adjusted by somatosensory feedback. The influence of feedback on cycle and phase durations is reduced by presynaptic inhibition, securing the RGs' operation in the flexor-driven regime. In the spinal-transected model (lacking supraspinal drives and presynaptic inhibition of somatosensory feedback), locomotion is entirely defined by sensory feedback from the hindlimbs, operating in a state-machine regime. Removing supraspinal drives on one side to simulate a hemisection enhances the gains of somatosensory feedback and its dominant role in controlling the timing of the transition between the flexion and extension states of the ipsilesional RG.

Modeling a lateral spinal hemisection represents an interesting and unique case. In the hemisected model shown in *Figure 1B*, the left and right spinal circuits still interact centrally as in the intact model, but the left contralesional circuits are mostly controlled by supraspinal drives, with the left RG mostly operating in a flexor-driven regime. On the right ipsilesional side, the circuits are mainly controlled by somatosensory feedback, with the right RG operating in a state-machine regime.

### Simulation of tied-belt treadmill locomotion in hemisected cats

We used the hemisected model (*Figure 1B*) to investigate speed-dependent changes of cycle and phase durations in both the ipsilesional (hemisected) and contralesional (intact) sides in the tied-belt

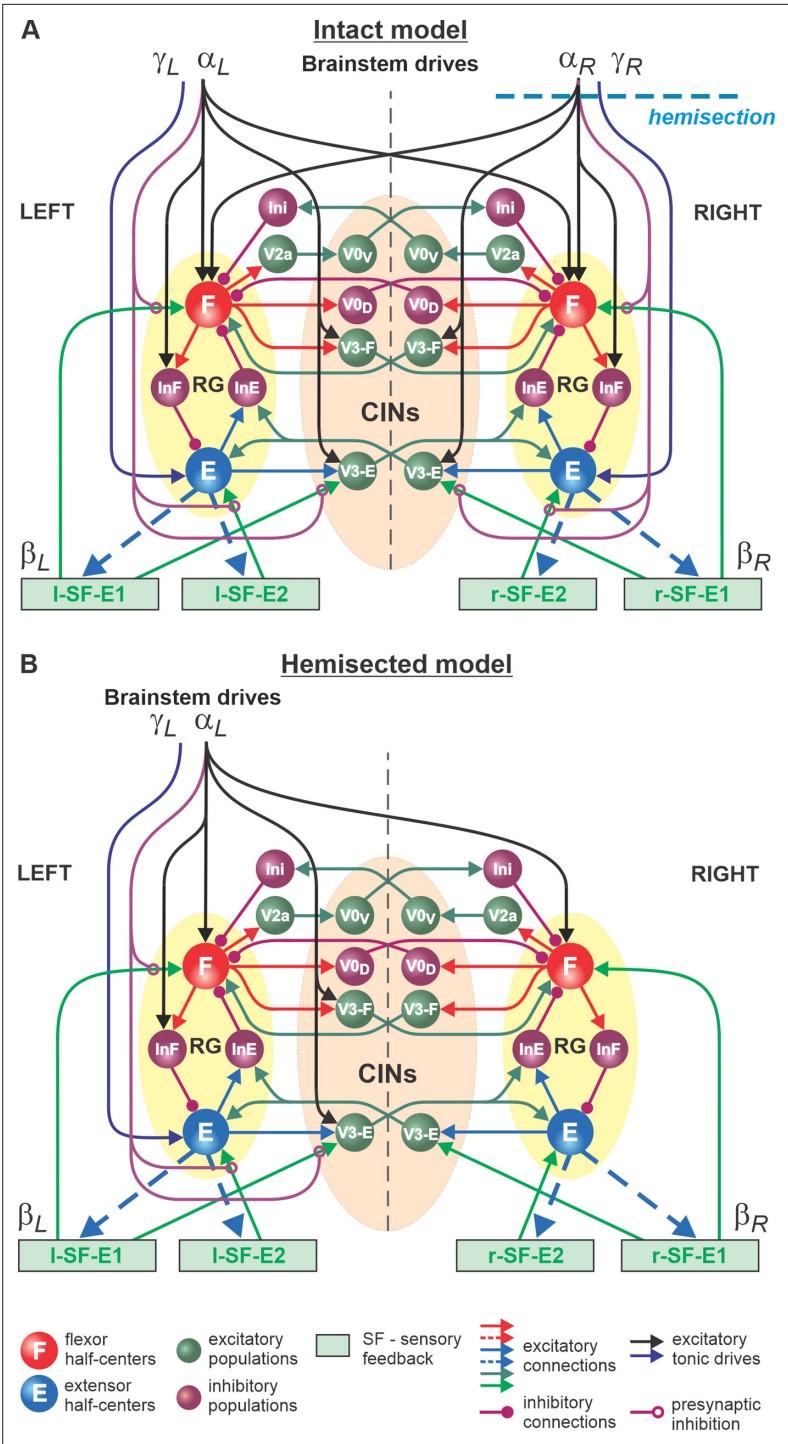

**Figure 1.** Model of spinal circuits controlling treadmill locomotion. (**A**) Model of the intact system ('intact model'). The model includes two bilaterally located (left and right) rhythm generators (RGs) (each is similar to that shown in **Rybak et al., 2024**, **Figure 3C**) coupled by (interacting via) several commissural pathways mediated by genetically identified commissural ($V0_D$, $V0_V$, and V3) and ipsilaterally projecting excitatory (V2a) and inhibitory neurons (see text for details). Left and right excitatory supraspinal drives $\left(\alpha_L \text{ and } \alpha_R\right)$ provide activation for the flexor half-centers (F) of the RGs (ipsi- and contralaterally) and some interneuron populations in the model, as well as for the extensor half-centers (E) ($\gamma_L$ and $\gamma_R$ ipsilaterally). Two types of feedback (SF-E1 and SF-E2) operating during extensor phases affect (excite), respectively, the ipsilateral F (SF-E1) and E (SF-E2) half-centers, and through V3-E neurons affect contralateral RGs. The SF-E1 feedback depends on the speed of the ipsilateral 'belt' $\left(\beta_L \text{ or } \beta_R\right)$

*Figure 1 continued on next page*

*Figure 1 continued*

and contributes to the extension-to-flexion transition on the ipsilateral side. The SF-E2 feedback activates the ipsilateral E half-center and contributes to 'weight support' on the ipsilateral side. The ipsilateral excitatory drives ($\alpha_L$ and $\alpha_R$) suppress the effects of all ipsilateral feedback inputs by presynaptic inhibition. The detailed description and all model parameters can be found in the preceding paper (*Rybak et al., 2024*). (**B**) Model of the right-hemisected system. All supraspinal drives on the right side (and their suppression of sensory feedback from the right limb) are eliminated from the schematic shown in (**A**).

condition. We simulated an increase in treadmill speed by progressively increasing the parameters $\beta_L = \beta_R$ that characterize the speed of the left and right belts, respectively (*Rybak et al., 2024*). Similar to our modeling of intact cat locomotion (see *Rybak et al., 2024*), we assumed that despite a thoracic hemisection on the right side, cats can still voluntary adjust the left supraspinal drive, $\alpha_L$, to compensate for treadmill speed and maintain a fixed body position relative to the external space. In contrast, after hemisection, the right part of the spinal circuits, including the right RG, is mainly controlled by somatosensory feedback, as all right supraspinal drives and their presynaptic inhibition of somatosensory inputs are removed (*Figure 1B*). All parameters of the remaining pathways and neurons in the hemisected model were identical to the intact model parameters from *Rybak et al., 2024*.

*Figure 2A1 and A2* show changes in cycle and phase durations in the model on the left contralesional and right-hemisected sides, respectively, with an increase of treadmill speed ($\beta_L = \beta_R = 0.4–1.0$). The results of simulations can be compared to our previous simulations of tied-belt locomotion performed using the intact model (*Rybak et al., 2024*, Fig. 5A) as shown in *Figure 2B1 and B2*.

Based on this comparison, the following qualitative observations can be made for tied-belt locomotion. (1) The intact side of the hemisected model shows speed-dependent changes of cycle and phase durations that are almost identical to those of the intact model (*Figure 2B1*). (2) The lesioned side shows an increase in flexor phase durations and a decrease of extensor phase durations relative to the intact state without changes in cycle durations (*Figure 2B2*).

## Simulation of split-belt treadmill locomotion in hemisected cats

Speed-dependent control of cycle and phase durations during locomotion on a split-belt treadmill significantly depends on whether the intact or lesioned limb is stepping on the slow or fast belt (*Lecomte et al., 2022*). Note that, according to our concept (*Rybak et al., 2024*), the intact side is mostly controlled by the supraspinal drives, whereas the hemisected side is mainly controlled by somatosensory feedback. We followed our previous experimental and modeling studies, in which the speed of the slow belt was fixed at 0.4 m/s in experiments and $\beta = 0.4$ in the model, and the speed on the fast side progressively increased from 0.5 to 1.0 m/s in experiments and from $\beta = 0.5$ to $\beta = 1.0$ in the model.

### Left slow/Right fast condition

*Figure 3A1 and A2* show the results of our simulation using the hemisected model in the *Left slow/Right fast* condition, with the limb controlled by the left contralesional side of the network stepping on the slow belt ($\beta_L = 0.4$) and the limb controlled by the right ipsilesional side stepping on the fast belt at speeds increasing from $\beta_R = 0.5$ to $\beta_R = 1.0$. With an increase of $\beta_R$, cycle and phase durations of the left contralesional (slow) side remained relatively constant (*Figure 3A1*), whereas on the right-hemisected (fast) side, we observed a pronounced speed-dependent reduction of extensor phase durations compensated by a corresponding increase of flexor phase durations. In the right ipsilesional hindlimb, stance and swing phase durations intersected at $\beta_R = 0.6$-$0.7$ and, with a further increase of $\beta_R$, the duty factor became less than 50% (*Figure 3A2*).

By comparing speed-dependent changes in cycle and phase durations obtained from the intact model in our previous simulations (*Rybak et al., 2024*, Fig. 6A), we can see that on the left contralesional (slow) side, cycle and extensor phase durations are longer in comparison to the intact model without noticeable differences in flexor phase durations (*Figure 3*). The most dramatic changes relative to the intact model were seen on the right-hemisected (fast) side where a significant speed-dependent reduction of extension and corresponding increase of flexion occurred with a duty factor falling below 50% in the middle of the speed interval (*Figure 3B2*). These changes of locomotor characteristics on the hemisected side resemble those in the fully spinal-transected model (see *Rybak*

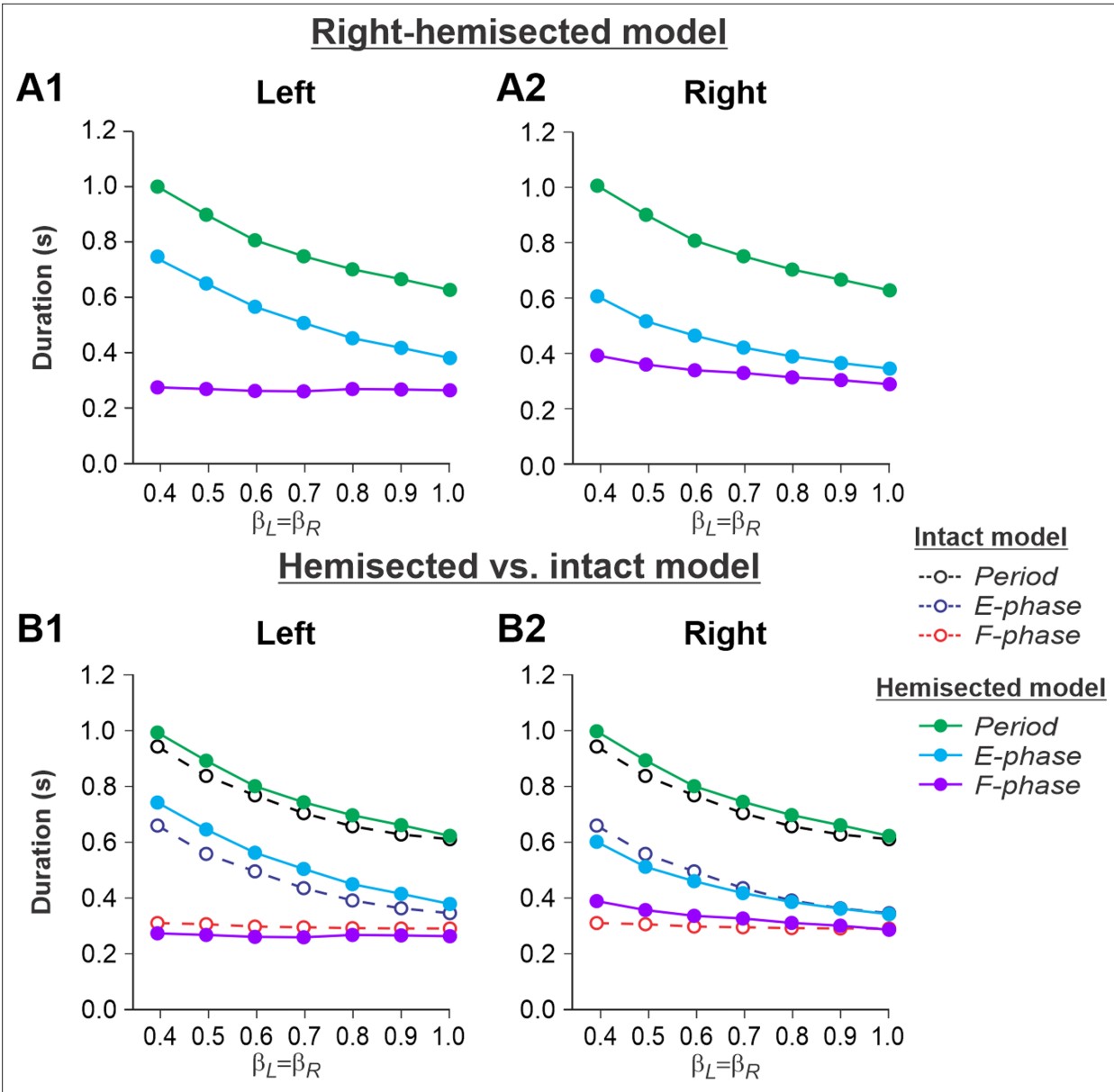

**Figure 2.** Simulation of locomotion on a tied-belt treadmill using the right-hemisected model. (**A1**, **A2**) Changes in the durations of cycle period and extensor/stance and flexor/swing phases during simulated tied-belt locomotion on the left contralesional (**A1**) and right ipsilesional (**A2**) sides of the model with an increasing simulated treadmill speed. (**B1**, **B2**) The curves from panels (**A1**) and (**A2**) for the hemisected model are superimposed with the corresponding curves obtained from the intact model (modified from *Rybak et al., 2024*, Fig. 5A).

*et al., 2024*, Fig. 6B, right). This is because the left-side drive is small at slow speeds and there is not much drive from the left side to the right side even though the latter is still receiving some supraspinal drive, as opposed to the full transection model.

### Left fast/Right slow conditions

*Figure 4A1 and A2* show the results of our simulation using the hemisected model in the *Left fast/Right slow* conditions, with the limb controlled by the left contralesional side of the network stepping on the fast belt ($\beta_L$ increasing from 0.5 to 1.0) and the limb controlled by the right ipsilesional side stepping on the slow belt ($\beta_R = 0.4$). In this case, cycle and phase durations on both sides do not change much with an increase of $\beta_L$ (cycle and extension durations slightly decrease on both sides and the flexion durations slightly increase on the intact side and slightly decrease on the hemisected side; *Figure 4A1 and A2*). Comparisons with the corresponding speed-dependent changes in cycle and

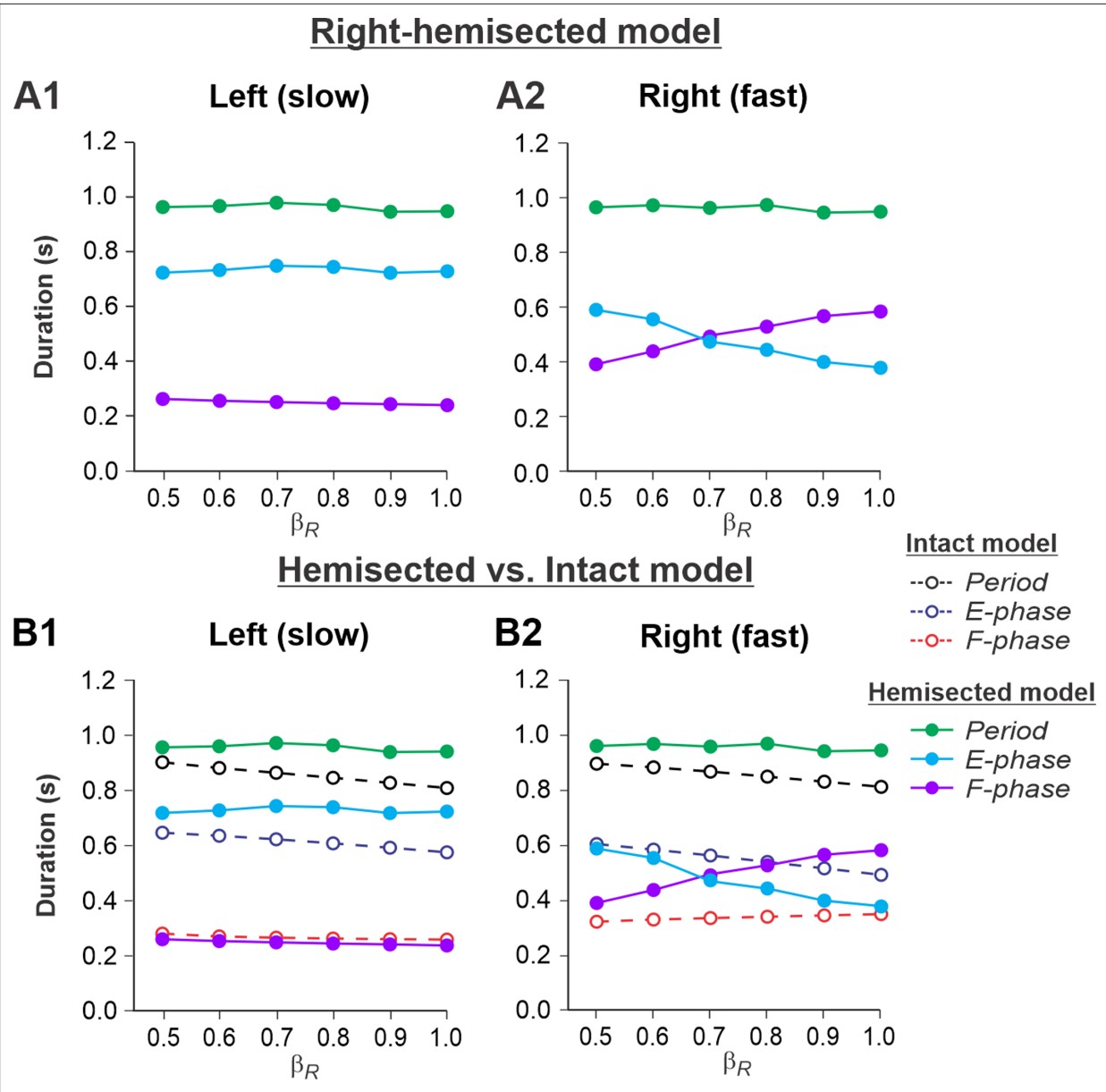

**Figure 3.** Simulation of locomotion on a split-belt treadmill using the right-hemisected model in the *Left slow/Right fast condition,* with the ipsilesional hindlimb stepping on the fast belt. (**A1, A2**) Changes in the durations of cycle period and extensor/stance and flexor/swing phases during simulated split-belt locomotion on the left contralesional (**A1**) and right ipsilesional (**A2**) sides of the model. (**B1, B2**) The curves from panels (**A1**) and (**A2**) for the hemisected model are superimposed with the corresponding curves obtained from the intact model (modified from *Rybak et al., 2024*, Fig. 5A).

phase durations in our intact model (*Rybak et al., 2024*, Fig. 6A) show similar changes for both fast and slow sides (*Figure 4B1 and B2*).

Based on our simulations, we formulated the following modeling predictions:

1. During tied-belt treadmill locomotion, speed-dependent changes of cycle and phase durations of the contralesional hindlimb are similar to those in the intact conditions, whereas the ipsilesional hindlimb shows noticeable changes in comparison with the intact conditions, such as shorter stance phases and longer swing phases.
2. During split-belt treadmill locomotion with the ipsilesional hindlimb stepping on the fast belt, cycle and stance durations of the contralesional hindlimb increase relative to the intact condition without noticeable changes in swing durations. At the same time, the most significant changes are observed on the ipsilesional side where the hindlimb is stepping on the fast belt. Specifically, in the right ipsilesional hindlimb, stance phases considerably decrease, and swing

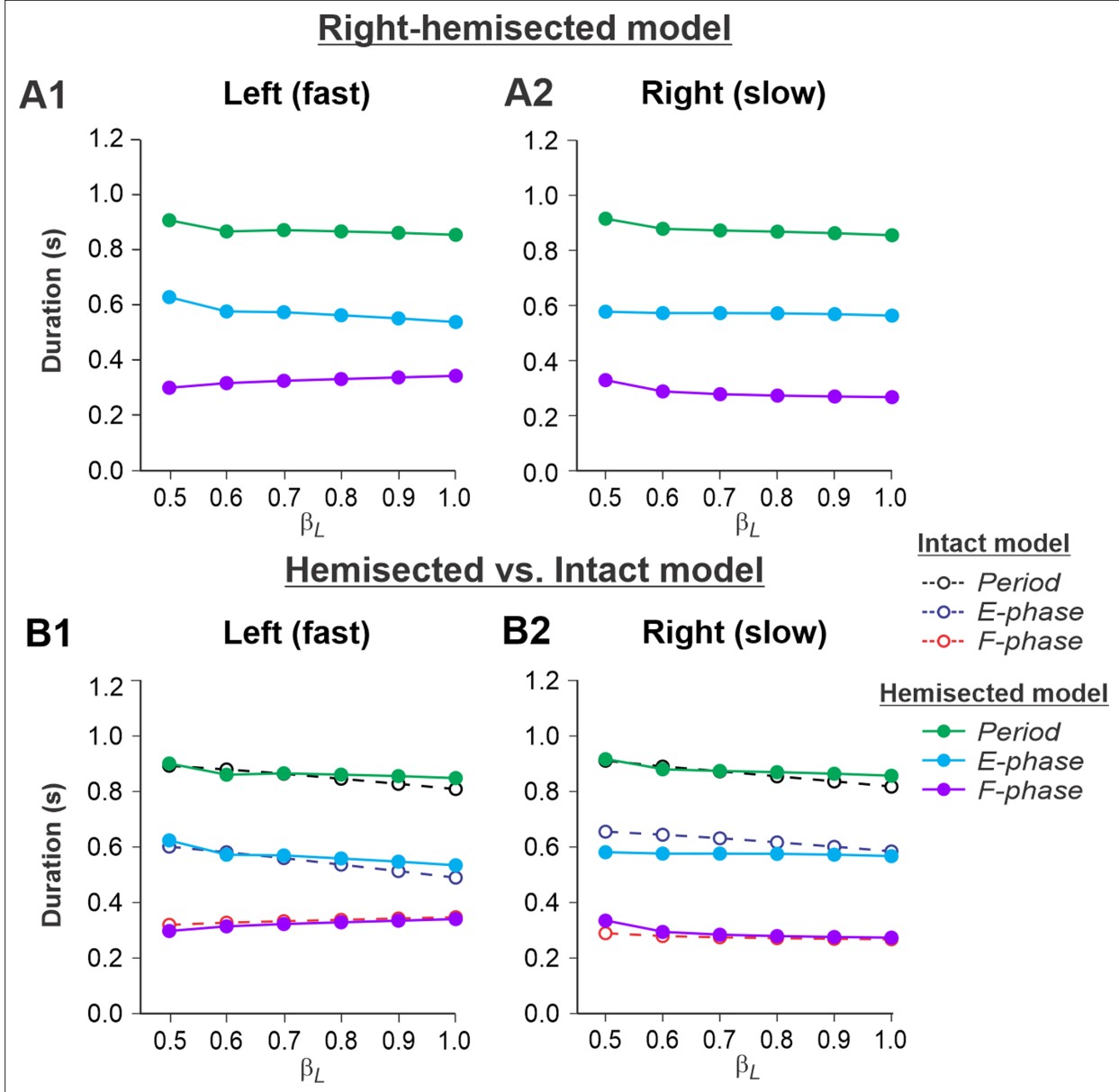

**Figure 4.** Simulation of locomotion on a split-belt treadmill using the right-hemisected model in the *Left fast/Right slow conditions,* with the ipsilesional hindlimb stepping on the slow belt. (**A1, A2**) Changes in the durations of locomotor period and extensor/stance and flexor/swing phases during simulated split-belt locomotion on the left contralesional (**A1**) and right ipsilesional (**A2**) sides of the model. (**B1, B2**) The curves from panels (**A1**) and (**A2**) for the hemisected model are superimposed with the corresponding curves obtained from the intact model (modified from *Rybak et al., 2024,* Fig. 5A).

phases increase with speed. As a result, swing durations begin to exceed stance durations (duty factor become less than 50%). Hence, having the ipsilesional hindlimb stepping on the fast belt makes locomotion more asymmetric (limping).

3. During split-belt treadmill locomotion with the ipsilesional hindlimb stepping on the slow belt, changes in cycle and phase durations in both hindlimbs with an increase of the speed of the fast belt are similar to those of intact cats. Hence, having the 'injured' hindlimb stepping on the slow belt makes locomotion more similar to locomotion of intact cats and more symmetrical, reducing limping.

# Studying the effects of thoracic lateral hemisection on cycle and phase durations during tied-belt and split-belt locomotion in cats

## Experimental procedures

We recorded cycle and phase durations of the left and right hindlimbs before and after a mid-thoracic (T5-T6) lateral hemisection on the right side in nine cats during tied-belt locomotion from 0.4 to 1.0 m/s, and in eight cats during split-belt locomotion with the slow side stepping at 0.4 m/s and the fast side stepping from 0.5 m/s to 1.0 m/s. For this study, we only retained for analysis cats that stepped up to 1.0 m/s during tied-belt (n = 5) and split-belt (n = 6) locomotion 7–8 weeks after hemisection. During split-belt locomotion, both the right ipsilesional (hemisected) and left contralesional limbs stepped on the slow or fast belts.

## Tied-belt locomotion

*Figure 5* shows speed-dependent changes in cycle, stance, and swing durations of both hindlimbs during tied-belt locomotion in cats before (intact, *Figure 5A and A2*) and 7–8 weeks after hemisection (*Figure 5B1 and B2*), as well as their comparisons (*Figure 5C1 and C2*). During tied-belt locomotion, cycle duration significantly decreased with increasing speed in the left ($p=1.81 \times 10^{-12}$) and right ($p=1.01 \times 10^{-12}$) hindlimbs with no significant difference between the intact (*Figure 5A1 and A2*) and hemisected (*Figure 5B1 and B2*) states ($p=0.074$ and $p=0.076$ for the left and right hindlimbs, respectively) (*Figure 5C1 and C2*). Stance phase durations also significantly decreased with increasing speed for the left ($p=1.07 \times 10^{-17}$) and right ($p=3.14 \times 10^{-18}$) hindlimbs. Stance duration was noticeably longer after hemisection for the left contralesional hindlimb ($p=0.045$), but not for the right ipsilesional hindlimb ($p=0.278$). Swing duration significantly decreased with increasing speed after hemisection for the left contralesional hindlimb ($p=0.011$) and the right ipsilesional hindlimb ($p=5.99 \times 10^{-5}$). Overall, the hemisection had no significant effect on swing duration for the left contralesional hindlimb compared to intact ($p=0.231$, *Figure 5C1*). However, in the right ipsilesional hindlimb, swing duration across speeds became significantly longer after hemisection compared to intact ($p=0.033$, *Figure 5C2*).

Experimental results were similar to simulated results (*Figure 5—figure supplement 1*) during tied-belt locomotion for the following. (1) On the contralesional left side of hemisected cats, speed-dependent changes of cycle and phase durations were qualitatively similar to those of intact cats. (2) On the ipsilesional side of hemisected cats, cycle and phase durations changed considerably, particularly because of a large proportional increase of the swing phase duration. However, in contrast to our simulations, the stance durations on the right ipsilesional side did not change following hemisection.

## Split-belt locomotion

Note that in our experiments the hemisection was always made on the right side. For split-belt locomotion, two condition types were applied, similar to that in the modeling: (1) *Left slow/Right fast* and (2) *left fast/Right slow* conditions, in which the right ipsilesional hindlimb stepped on the fast and slow belts, respectively. Slow always refers to a belt speed of 0.4 m/s while fast refers to belt speeds from 0.5 m/s to 1.0 m/s.

### Left slow/Right fast condition

The results of cycle and phase durations for the *Left slow/Right fast* condition are shown in *Figure 6* before (*Figure 6A1 and A2*) and 7–8 weeks after hemisection (*Figure 6B1 and B2*), as well as comparisons before and after hemisection (*Figure 6C1 and C2*).

Cycle duration slightly decreased with increasing right belt speed in the left ($p=1.2 \times 10^{-7}$) and right ($p=1.87 \times 10^{-7}$) hindlimbs. After hemisection, cycle duration across speeds was significantly greater compared to the intact state for the left ($p=0.044$) and right ($p=0.046$) hindlimbs. Additionally, after hemisection, stance durations across speeds were significantly greater compared to the intact state for the left side ($p=0.034$) without significant differences in swing durations on the same side ($p=0.096$). We found small but significant decreases in stance ($p=0.005$) and swing ($p=2.40 \times 10^{-11}$) durations for the left hindlimb with increasing right belt speed. The effects of speed were much more pronounced for the right hindlimb, where stance ($p=1.53 \times 10^{-15}$) and swing ($p=7.18 \times 10^{-9}$) durations significantly decreased and increased with an increase in right belt speed. The most noticeable changes after

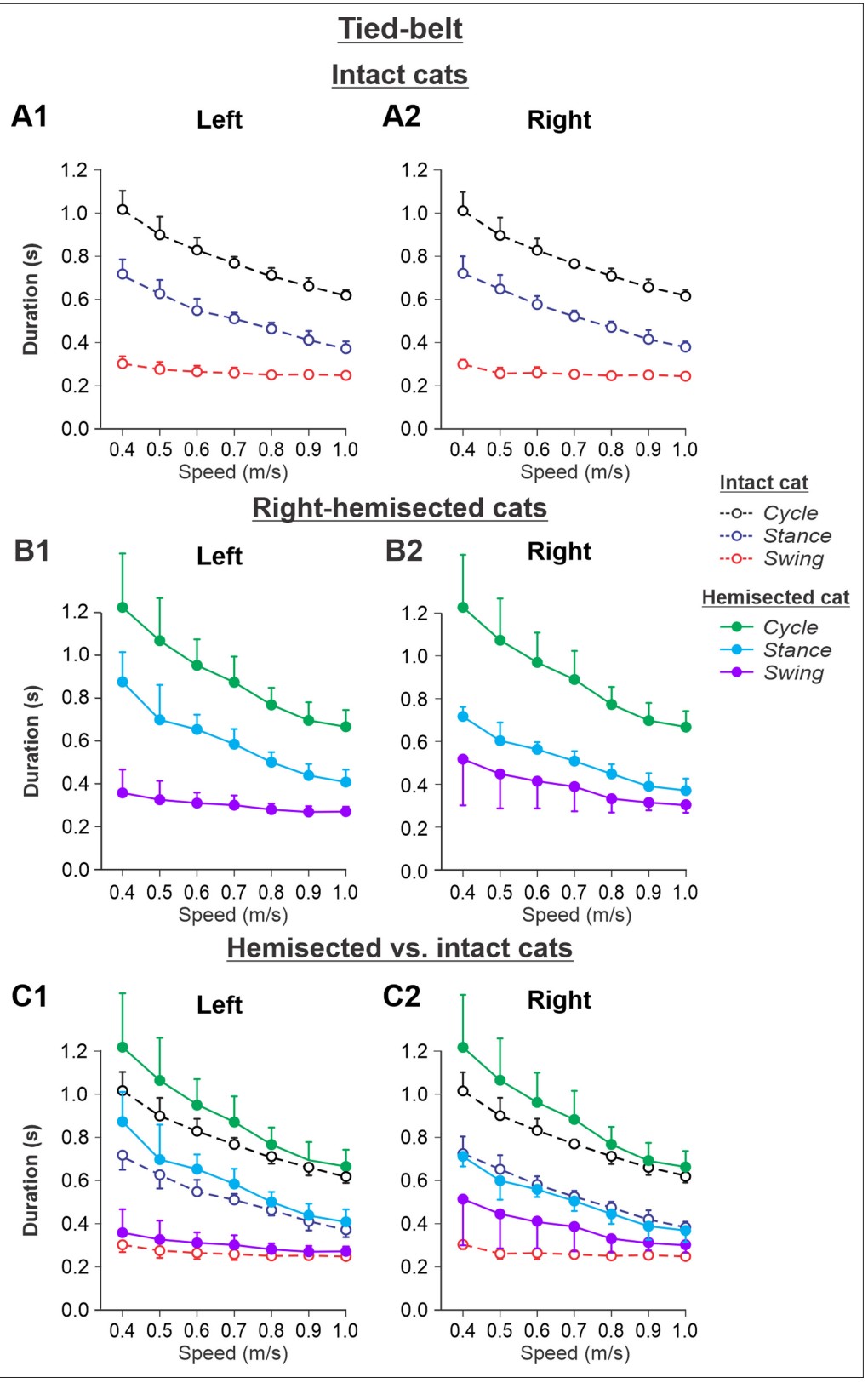

**Figure 5.** Cycle and phase durations before and after a right thoracic lateral hemisection in cats walking on a tied-belt treadmill. (**A1, A2**) Changes in the durations of cycle period and stance and swing phases during tied-belt locomotion in intact cats (before hemisection). (**B1, B2**) The same characteristics obtained from cats 7–8 weeks

*Figure 5 continued on next page*

*Figure 5 continued*
after a right lateral thoracic hemisection. (**C1, C2**) The curves from panels (**B1**) and (**B2**) for hemisected cats are superimposed (for comparison) with the corresponding curves (**A1, A2**) obtained from intact cats.

The online version of this article includes the following figure supplement(s) for figure 5:

**Figure supplement 1.** Comparison of simulated and experimental data during tied-belt locomotion.

hemisection occurred in the right ipsilesional hindlimb stepping on the fast belt. Swing (p=0.024) but not stance (p=0.161) durations were significantly greater after hemisection compared to the intact state. As a result, swing durations in the right ipsilesional hindlimb became longer than stance duration at speeds of 0.7 m/s and above (with a duty factor below 50%), which was not observed in intact cats.

Experimental results were similar to simulated results (*Figure 6—figure supplement 1*) in the Left slow/Right fast condition for the following. (1) In the left contralesional hindlimb, cycle and stance durations were significantly greater after hemisection compared to the intact state without significant changes in swing durations. (2) In the ipsilesional right hindlimb, stance durations significantly decreased while swing durations increased with an increase in right belt speed. Swing durations became longer than stance durations at a right belt speed of about 0.6–0.7 m/s. Thus, with the right ipsilesional hindlimb on the fast belt, the asymmetry of stance and swing durations between the left and right sides increased. In contrast to our simulations, however, the hemisection did not change the stance duration in the ipsilesional hindlimb relative to the intact state, although swing durations increased, similar to our simulations.

## Left fast/Right slow conditions

The results of cycle and phase durations for the *Left fast/Right slow* conditions are shown in *Figure 7* before (*Figure 7A1 and A2*) and 7–8 weeks after hemisection (*Figure 7B1 and B2*), as well as comparisons before and after hemisection (*Figure 7C1 and C2*).

Cycle duration significantly decreased with increasing left belt speed in the left (p=4.41 × 10$^{-11}$) and right (p=3.83 × 10$^{-5}$) hindlimbs, with no significant difference between the intact and hemisected states (p=0.283 and p=0.212 for the left and right hindlimbs, respectively). Stance duration significantly decreased with increasing left belt speed in the left hindlimb (p=1.02 × 10$^{-18}$) but not in the right hindlimb (p=0.131), with no effect of the hemisection (p=0.066 and p=0.996 for the left and right hindlimbs, respectively). Swing duration significantly increased with increasing left belt speed in the left hindlimb (p=0.010) while it decreased in the right hindlimb (p=3.79 × 10$^{-10}$). Swing duration was significantly longer after hemisection in the right hindlimb (p=0.003) but not in the left hindlimb (p=0.985).

The experimental results generally confirm our modeling predictions (*Figure 7—figure supplement 1*) that spinal hemisection does not change cycle and phase durations on both sides in comparison with the corresponding characteristics in intact cats when the ipsilesional hindlimb is stepping on the slow belt, other than a small increase of swing duration in the ipsilesional hindlimb.

## Discussion

The goal of this study was to investigate the effects of a lateral thoracic hemisection on the operation of spinal sensorimotor circuits controlling the main temporal locomotor characteristics of cats walking on a tied-belt and split-belt treadmill over a range of speeds and left-right speed differences. We employed a combination of computational modeling and experimental approaches. We used our previously published model (*Rybak et al., 2024*) that described and highlighted the specific changes in the roles of supraspinal drives and somatosensory feedback following spinal transections, and hypothesized that after a lateral hemisection the contralesional ('intact') side of the spinal network is mostly controlled by supraspinal drives, whereas the ipsilesional ('injured') side is mostly controlled by somatosensory feedback. Using this network architecture and proposed operating regimes, we simulated tied-belt and split-belt treadmill locomotion following a right thoracic lateral hemisection and computed cycle and phase durations. We then compared the simulation results with those obtained during experiments in adult cats, before and after a mid-thoracic hemisection on the right side in the same locomotor conditions. The simulated and experimental results concerning hemisection-produced

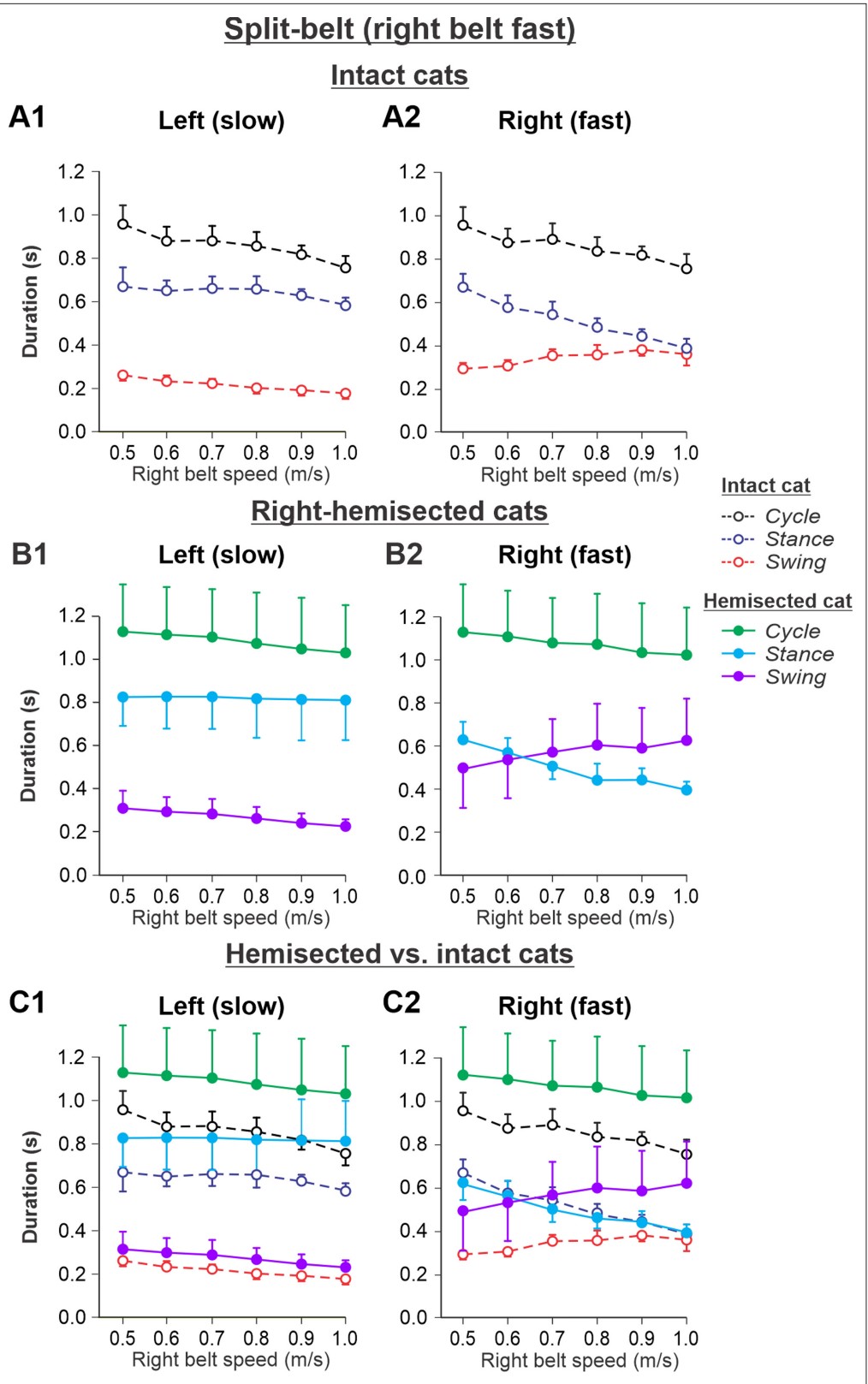

**Figure 6.** Cycle and phase durations before and after a right thoracic lateral hemisection in cats walking on a split-belt treadmill in the *Left slow/Right fast condition,* with the ipsilesional hindlimb stepping on the fast belt. (**A1, A2**) Changes in the durations of cycle period and stance and swing phases during split-belt locomotion in intact cats (before hemisection). (**B1, B2**) The same characteristics obtained from cats 7–8 weeks after a right lateral thoracic

*Figure 6 continued*

hemisection. (**C1, C2**) The curves from panels (**B1**) and (**B2**) for injured cats are superimposed (for comparison) with the corresponding curves (**A1, A2**) obtained from intact cats.

The online version of this article includes the following figure supplement(s) for figure 6:

**Figure supplement 1.** Comparison of simulated and experimental data during split-belt locomotion in the Left slow/Right fast condition.

changes in cycle, swing, and stance durations during tied-belt and split-belt treadmill locomotion across speeds were qualitatively similar. The following sections discuss the similarities and differences between simulated and experimental results, what these mean for understanding possible changes in the operation of spinal sensorimotor circuits after incomplete SCI, as well as limitations of the model and future directions.

## Spinal sensorimotor network architecture and operation after a lateral spinal hemisection

Although our hemisected model is built on many simplifications, as discussed in detail in *Rybak et al., 2024*, it generated cycle and phase durations that qualitatively matched those obtained experimentally during tied-belt and split-belt locomotion. This provides additional validation of our model as it can reproduce not only experimental data in intact and spinal-transected animals (*Rybak et al., 2024*), but also, without additional changes or adjustments of model parameters, it makes several predictions on expected changes in locomotor characteristics following an incomplete SCI.

During tied-belt locomotion, the model predicted a decrease in cycle and stance durations with increasing speed following a right thoracic lateral hemisection, similar to the intact state (*Figure 2*), which we also observed experimentally (*Figure 5*). In both simulated and experimental results, swing duration increased on the ipsilesional side following hemisection compared to the intact state and was longer compared to the contralesional left side. Stance duration was also slightly longer after hemisection in the contralesional left hindlimb in both simulated and experimental results. Based on our model (*Figure 1*), the increase of ipsilesional swing duration is because of a greater influence of feedback from muscle afferents of right hip flexors (r-SF-E1) on the right flexor half-center due to the loss of presynaptic inhibition from supraspinal drive. The increase in stance duration on the contralesional left side, according to our model, could be due to increased inhibition from the right flexor half-center via commissural interneurons. In cats, we cannot exclude that they can use a voluntary strategy to shift weight support to the less affected hindlimb.

During split-belt locomotion in the Left slow/Right fast condition, the hemisected model predicted longer swing durations in the right ipsilesional hindlimb that became longer than stance duration at around 0.7 $\beta_R$ (*Figure 3*), which we also observed experimentally at a corresponding right belt speed of 0.7 m/s (*Figure 6*). Again, according to the model, this is due to the release from inhibition of the r-SF-E1 feedback and its excitatory influence on the right flexor half-center. The model also correctly predicted the increased stance phase duration in the left contralesional hindlimb after the hemisection, which based on the model may correspond to greater crossed inhibition of the left flexor half-center from the right flexor half-center. The stance duration of the left hindlimb stepping on the slow belt also increased compared to the right hindlimb stepping on the fast belt. This asymmetry in phase durations also generates an asymmetry in somatosensory feedback, particularly load-related feedback. Greater load-related group Ib feedback (l-SF-E2 in the model) for the left slow limb in the Left slow/Right fast condition increases excitatory inputs to the left extensor half- center, prolonging left hindlimb stance duration. In cats, we observe a greater body shift toward the slow belt and greater extensor activity in muscles of the limb stepping on the slow belt, consistent with greater group Ib feedback (*Frigon et al., 2015*; *Park et al., 2019*). In the Left fast/Right slow condition, the model predicted almost equal stance and swing durations for the left contralesional and right ipsilesional hindlimbs after hemisection compared to the intact model, with mainly a decrease in stance duration of the left hindlimb as speed of the left belt increased (*Figure 4*). This is qualitatively similar to what we observed experimentally (*Figure 7*). In summary, our computational model was able to predict and explain several experimental results.

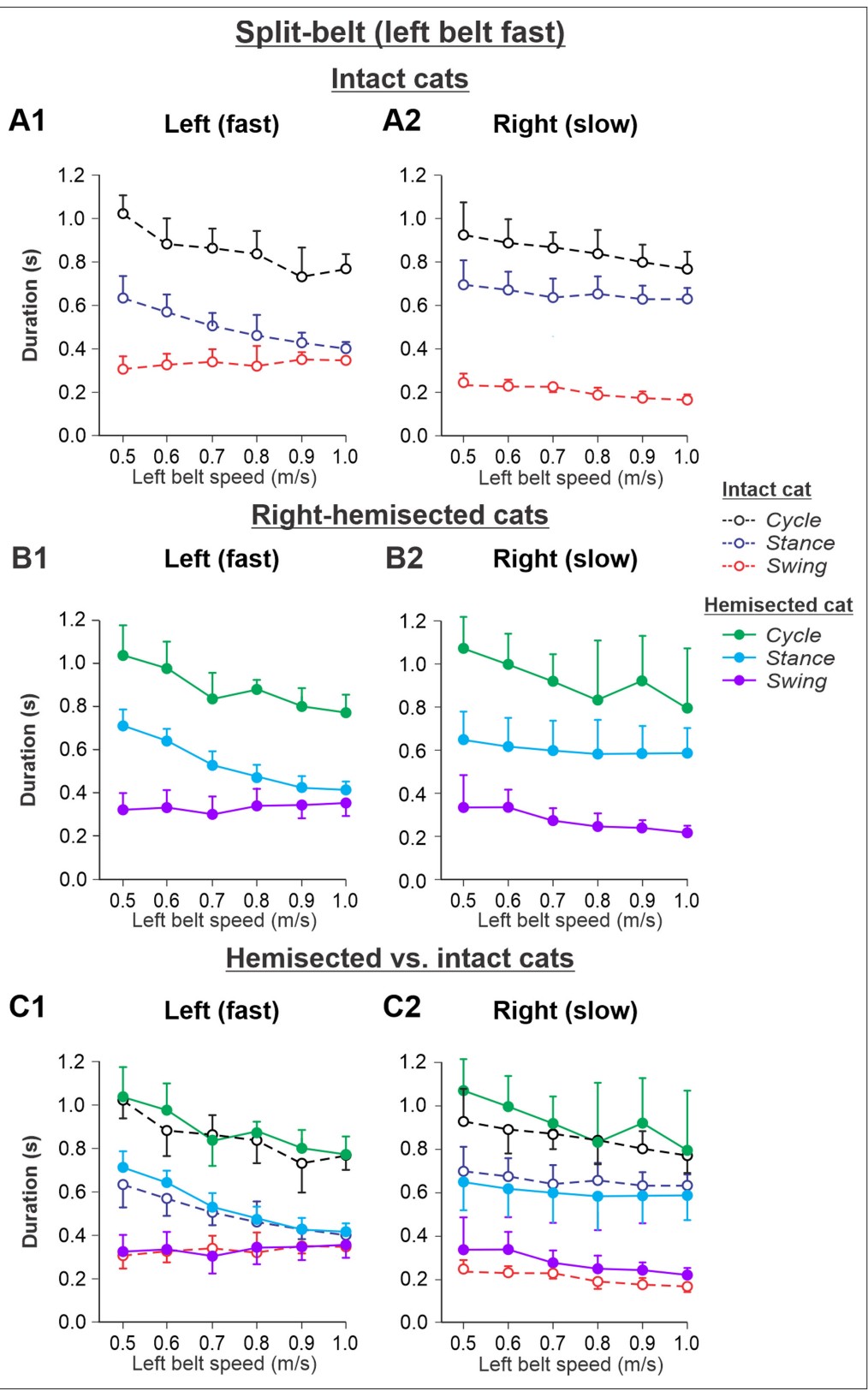

**Figure 7.** Cycle and phase durations before and after a right thoracic lateral hemisection in cats walking on a split-belt treadmill in the *Left fast/Right slow condition,* with the ipsilesional hindlimb stepping on the slow belt. (**A1, A2**) Changes in the durations of cycle period and stance and swing phases during split-belt locomotion in intact cats (before hemisection). (**B1, B2**) The same characteristics obtained from cats 7–8 weeks after a right

*Figure 7 continued on next page*

*Figure 7 continued*

lateral thoracic hemisection. (**C1, C2**) The curves from panels (**B1**) and (**B2**) for injured cats are superimposed (for comparison) with the corresponding curves (**A1, A2**) obtained from intact cats.

The online version of this article includes the following figure supplement(s) for figure 7:

**Figure supplement 1.** Comparison of simulated and experimental data during split-belt locomotion in the *Left fast/Right slow condition.*

## Functional considerations

What we observed, with both simulated and experimental results, is that having the ipsilesional hindlimb stepping on the fast belt increased the asymmetry between hindlimbs compared to the intact state, with longer swing and stance durations for the ipsilesional and contralesional hindlimbs, respectively, in the Left slow/Right fast split-belt condition. In other words, having the ipsilesional hindlimb step on the fast belt increased the left-right asymmetry observed during split-belt locomotion. However, having the ipsilesional hindlimb step on the slow belt in the Left fast/Right slow split-belt condition produced simulation and experimental results that were similar to the intact state. Why does having the ipsilesional hindlimb step on the slow belt reduce the left-right asymmetry generated by the lateral hemisection? We propose that it forces the ipsilesional hindlimb to increase its support as swing duration of the left contralesional hindlimb increases as left belt speed also increases. This increases the excitatory influence of r-SF-E2 or group Ib feedback on the right extensor half-center. Without this increase in the support of the ipsilesional right hindlimb stepping on the slow belt, the animal could not perform split-belt locomotion. From a clinical perspective, prolonged use of split-belt locomotion for rehabilitation might be a way to reduce asymmetries following lateralized neurological injuries, such as SCIs and stroke (*Reisman et al., 2010*; *Reisman et al., 2013*; *Kuczynski et al., 2017*; *Lecomte et al., 2022*).

## Limitations and future directions

The network architecture and sensorimotor interactions of the model are obviously based on many simplifications, as described in *Rybak et al., 2024*. For instance, the model does not account for the interactions between the fore- and hindlimbs (*Frigon, 2017*; *Harnie et al., 2024*) and does not include properties of the musculoskeletal system and a pattern formation network that shape neural outputs (*McCrea and Rybak, 2008*; *Markin et al., 2016*; *Frigon et al., 2021*). The lack of possible plastic changes in spinal sensorimotor circuits of our model may explain the absence of exact/quantitative correspondences between simulated and experimental data. In future computational studies designed to explain post-injury changes, it will also be important to consider neuroplastic changes that occur over time following SCI, such as changes in neuronal excitability and the establishment of new neural pathways (*Raineteau and Schwab, 2001*; *Bareyre et al., 2004*; *Murray et al., 2010*; *Rossignol and Frigon, 2011*). In the present study, we only compared simulated results to experimental results obtained 7–8 weeks after hemisection, when locomotor recovery was relatively robust. Experimentally, it is also difficult to surgically perform a perfect lateral hemisection and some spinal tissue is often damaged on the contralateral side and/or spared on the ipsilesional side. Nevertheless, although some characteristic changes predicted by the model were not fully consistent with our experimental results, we observed striking similarities between our simulations and the consequent experimental studies. This indicates that modeling sensorimotor interactions can be a powerful tool to predict and mechanistically explain changes following complete SCI, possibly serving as tool to guide rehabilitation efforts.

## Materials and methods
### Modeling

The detailed description of our model used in this study and the full set of parameters can be found in our recent *eLife* publication (*Rybak et al., 2024*). To simulate a right lateral hemisection, all supraspinal drives from the right side were set to zero, transforming the intact model (*Figure 1A*) to a hemisected model (*Figure 1B*). No other model parameters were changed. The simulation package NSM

2.5.7, the model configuration file necessary to create and run simulations, and the custom MATLAB scripts are available at https://github.com/RybakLab/nsm (copy archived at *RybakLab, 2024*).

## Experimental materials and methods

### Ethical approval

All experimental procedures were approved by the Animal Care Committee of the Université de Sherbrooke (Protocol 442-18) in accordance with policies and directives of the Canadian Council on Animal Care. We obtained the current data set from nine adult purpose-bred cats (>1 year of age at the time of experimentation), five females and four males, weighing between 4.0 kg and 6.2 kg, purchased from Marshall BioResources (North Rose, NY). Before and after experiments, cats were housed and fed (weight-dependent metabolic diet and water ad libitum) in a dedicated room within the animal care facility of the Faculty of Medicine and Health Sciences at the Université de Sherbrooke. We followed the ARRIVE guidelines 2.0 for animal studies (*Percie du Sert et al., 2020*). The investigators understand the ethical principles under which the journal operates, and our work complies with this animal ethics checklist. In order to maximize the scientific output of each animal, they were used in other studies to investigate different scientific questions, some of which have been published (*Lecomte et al., 2022*; *Merlet et al., 2022*; *Lecomte et al., 2023*; *Mari et al., 2023*; *Audet et al., 2023*; *Mari et al., 2024a*; *Mari et al., 2024b*).

### Surgical procedures

After collecting data in the intact state, we performed a lateral hemisection between the fifth and sixth thoracic vertebrae (T5-T6) on the right side of the spinal cord. Before surgery, we sedated the cat with an intramuscular injection of a cocktail containing butorphanol (0.4 mg/kg), acepromazine (0.1 mg/kg), and glycopyrrolate (0.01 mg/kg) and inducted with another intramuscular injection of ketamine (2.0 mg/kg) and diazepam (0.25 mg/kg) in a 1:1 ratio. We shaved the fur overlying the back and the skin was cleaned with chlorhexidine soap. The cat was then anesthetized with isoflurane (1.5–3%) and $O_2$ using a mask for a minimum of 5 min and then intubated with a flexible endotracheal tube. Isoflurane concentration was confirmed and adjusted throughout the surgery by monitoring cardiac and respiratory rates, applying pressure to the paw to detect limb withdrawal, and assessing muscle tone. Once the animal was deeply anesthetized, an incision of the skin over and between the fifth and sixth thoracic vertebrae (T5-T6) was made and after carefully setting aside muscle and connective tissue, a small laminectomy of the corresponding dorsal bone was performed. Lidocaine (xylocaine, 2%) was applied topically followed by 2–3 intraspinal injections on the right side of the cord. We then sectioned the spinal cord laterally from the midline to the right using surgical scissors. We placed hemostatic material (Spongostan) within the gap before sewing back muscles and skin in anatomical layers. In the days following hemisection, voluntary bodily functions were carefully monitored. The bladder and large intestine were manually expressed if needed. At the end of surgery, we injected subcutaneously an antibiotic (cefovecin, 8 mg/kg) and a fast-acting analgesic (buprenorphine, 0.01 mg/kg). We also taped a fentanyl (25 μg/hr) patch to the back of the animal 2-3 cm rostral to the base of the tail for prolonged analgesia, which we removed 4–5 days later. After surgery, the cats were placed in an incubator and closely monitored until they regained consciousness. We administered another dose of buprenorphine ~7 hr after surgery. At the end of experiments, cats were anaesthetized with isoflurane (1.5–3.0%) and $O_2$ before receiving a lethal dose of pentobarbital (120 mg/kg) through the left or right cephalic vein. Cardiac arrest was confirmed using a stethoscope to determine the death of the animal. Spinal cords were then harvested for histological analysis (*Lecomte et al., 2022*; *Lecomte et al., 2023*).

### Experimental design and data collection

We collected EMG and kinematic data before and at different time points after the thoracic lateral hemisection during quadrupedal locomotion on a treadmill consisting of two independently controlled belts 130 cm long and 30 cm wide (Bertec) with a Plexiglas separator (130 cm long, 7 cm high, and 1.3 cm wide) placed between the two belts to prevent the limbs from impeding each other. In the intact preoperative state, cats were trained for 2–3 weeks in a progressive manner, first for a few steps and then for several consecutive minutes, using food and affection as reward. Once cats could perform 3–4 consecutive minutes of stepping, we started the experiments. For these experiments,

cats stepped in a tied-belt condition from 0.4 m/s to 1.0 m/s and in a split-belt locomotion condition with the slow belt at 0.4 m/s and the fast belt going from 0.5 m/s to 1.0 m/s, with speed increments of 0.1 m/s. Both the left and right sides were used on the slow and fast belts.

During experiments, two cameras (Basler AcA640-100gm) captured videos from the left and right sides of the animal (60 frames per second; 640 × 480 pixels spatial resolution). A custom-made LabVIEW program acquired the images. We analyzed kinematic data from videos off-line with a deep learning approach (DeepLabCut; *Mathis et al., 2018*), which we recently validated in our cat model (*Lecomte et al., 2021*). We determined contact of the left and right hindlimbs by visual inspection. Paw contact was defined as the first frame where the paw made visible contact with the treadmill surface. We measured cycle duration from successive paw contacts, while stance duration corresponded to the interval of time from foot contact to the most caudal displacement of the toe relative to the hip (*Halbertsma, 1983*). We calculated swing duration as cycle duration minus stance duration.

## Histology

After confirming euthanasia (i.e., no cardiac and respiratory functions), we harvested an approximately 2 cm long section of the spinal cord centered on the lesion. Segments of the dissected spinal cord were then placed in a 25 ml 4% paraformaldehyde solution (in 0.1 M phosphate-buffered saline [PBS], 4°C). After 5 days, we placed the spinal cord in a new PBS solution containing 30% sucrose for 72 hr at 4°C, then froze it in isopentane at –50°C for cryoprotection. The spinal cord was then sliced in 50 µm coronal sections using a cryostat (Leica CM1860, Leica BioSystems Inc, Concord, ON, Canada) and mounted on gelatinized-coated slides. The slides were dried overnight and then stained with a 1% cresyl violet acetate solution for 12 min. We washed the slides for 3 min in distilled water before being dehydrated in successive baths of ethanol (50%, 70%, and 100%, 5 min each) and transferring them in xylene for 5 min. Dibutylphthalate polystyrene xylene was next used to mount and dry the spinal cord slides before being scanned by a Nanozoomer 2.0-RS (Hamamatsu Corp, Bridgewater, NJ). We then performed qualitative and quantitative analyses to estimate lesion extent using ImageJ by selecting the slide with the greatest identifiable damaged area. Using the scarring tissue stained with cresyl violet acetate, we estimated lesion extent by dividing the lesion area by the total area of the selected slice and expressed it as percentage. Lesion extent estimations for individual cats after the right lateral hemisection ranged from 40.7% to 66.4% (50.3 ± 10.3%) (*Lecomte et al., 2023*).

## Statistical analysis

We performed statistical analyses using IBM SPSS Statistics 22.0 software. To determine the effect of locomotor state and speed on cycle, stance, and swing durations during quadrupedal tied- and split-belt locomotion, we performed a two-factor (state × speed) repeated-measure ANOVA for each dependent variable. Significance level was set at $p < 0.05$.

---

# Additional information

### Funding

| Funder | Grant reference number | Author |
| --- | --- | --- |
| National Institutes of Health | R01 NS110550 | Ilya A Rybak<br>Boris I Prilutsky<br>Alain Frigon |
| National Science Foundation | 2024414 | Boris I Prilutsky |

The funders had no role in study design, data collection and interpretation, or the decision to submit the work for publication.

### Author contributions

Ilya A Rybak, Conceptualization, Resources, Data curation, Software, Formal analysis, Supervision, Funding acquisition, Validation, Investigation, Visualization, Methodology, Writing – original draft, Project administration, Writing – review and editing; Natalia A Shevtsova, Conceptualization, Data

curation, Formal analysis, Validation, Visualization, Writing – original draft, Writing – review and editing; Johannie Audet, Sirine Yassine, Formal analysis, Investigation, Writing – review and editing; Sergey N Markin, Formal analysis, Validation, Writing – review and editing; Boris I Prilutsky, Conceptualization, Funding acquisition, Project administration, Writing – review and editing; Alain Frigon, Conceptualization, Resources, Data curation, Supervision, Funding acquisition, Writing – original draft, Project administration, Writing – review and editing

## Author ORCIDs
Ilya A Rybak https://orcid.org/0000-0003-3461-349X
Natalia A Shevtsova https://orcid.org/0000-0002-1971-9707
Boris I Prilutsky http://orcid.org/0000-0003-0499-3890
Alain Frigon https://orcid.org/0000-0002-9259-2706

## Ethics

All experimental procedures were approved by the Animal Care Committee of the Université de Sherbrooke (Protocol 442-18) in strict accordance with policies and directives of the Canadian Council on Animal Care. We obtained the current data set from 9 adult purpose-bred cats (> 1 year of age at the time of experimentation), 5 females and 4 males, weighing between 4.0 kg and 6.2 kg, purchased from Marshall BioResources (North Rose, NY, USA). Before and after experiments, cats were housed and fed (weight-dependent metabolic diet and water ad libitum) in a dedicated room within the animal care facility of the Faculty of Medicine and Health Sciences at the Université de Sherbrooke. We followed the ARRIVE guidelines 2.0 for animal studies (Percie du Sert et al., 2020). The investigators understand the ethical principles under which the journal operates, and our work complies with this animal ethics checklist. In order to maximize the scientific output of each animal, they were used in other studies to investigate different scientific questions, some of which have been published (Lecomte et al., 2022; Merlet et al., 2022; Lecomte et al., 2023; Mari et al., 2023; Audet et al., 2023; Mari et al., 2024a; Mari et al., 2024b).

Reviewer #1 (Public review): https://doi.org/10.7554/eLife.103504.3.sa1
Reviewer #2 (Public review): https://doi.org/10.7554/eLife.103504.3.sa2
Author response https://doi.org/10.7554/eLife.103504.3.sa3

---

# Additional files

## Supplementary files
MDAR checklist

## Data availability
Part of the current manuscript is a computational study. The simulation package used, the model configuration file necessary to create and run simulations, and the custom Matlab scripts are available at https://github.com/RybakLab/nsm (copy archived at *RybakLab, 2024*). The experimental data is available on Dryad at https://doi.org/10.5061/dryad.bk3j9kdp4.

The following dataset was generated:

| Author(s) | Year | Dataset title | Dataset URL | Database and Identifier |
|---|---|---|---|---|
| Frigon A | 2025 | Experimental data for eLife paper | https://doi.org/10.5061/dryad.bk3j9kdp4 | Dryad Digital Repository, 10.5061/dryad.bk3j9kdp4 |

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
