## [Editor Report · eLife Assessment]

This **important** modeling study alters a previous model of the intact cat spinal locomotor network to simulate a lateral hemi-section of the spinal cord. The modeling and experimental work described provide **convincing** evidence that this model is capable of qualitatively predicting alterations to the swing and stance phase durations during locomotion at different speeds on intact or split-belt treadmills. This paperarticle will interest neuroscientists studying vertebrate motor systems, including researchers working on motor dysfunction after spinal cord injury.

---

## [Referee Report · Reviewer #1 (Public review)]

Summary:

This study adapts a previously published model of the cat spinal locomotor network to make predictions of how phase durations of swing and stance at different treadmill speeds in tied-belt and split-belt conditions would be altered following a lateral hemisection. The simulations make several predictions that are replicated in experimental settings. This updated manuscript addressed well many of the reviewer comments made to the first version.

Strengths:

-Despite only altering the connections in the model, the model is able to replicate very well several experimental findings. This provides strong validation for the model and highlights its utility as a tool to investigate the operations of mammalian spinal locomotor networks.

-The study provides insights about interactions between the left and right side of the spinal locomotor networks, and how these interactions depend on the mode of operation, as determined by speed and state of the nervous system.

-The writing is logical, clear and easy to follow.

Comments on revisions:

My concerns were well addressed by the authors. I have no additional concerns

---

## [Referee Report · Reviewer #2 (Public review)]

This is a nice article that presents interesting findings. The model's predictions match the data, which is good. The discussion points to modeling plasticity after SCI, which will be important.

The manuscript is well-written and interesting, and the putative neural circuit mechanisms that the model uncovers are super cool if they can be tested in an animal.

---

## [Author Response]

The following is the authors’ response to the original reviews.

**eLife Assessment**
The modeling and experimental work described provide solid evidence that this model is capable of qualitatively predicting alterations to the swing and stance phase durations during locomotion at different speeds on intact or split-belt treadmills, but a revision of the figures to overlay the model predictions with the experimental data would facilitate the assessment of this qualitative agreement. This paper will interest neuroscientists studying vertebrate motor systems, including researchers investigating motor dysfunction after spinal cord injury.

Figures showing the overlay of the experimental data with the modeling predictions have been included as figure supplements for Figures 5-7. This highlights how accurate the model predictions were.

**Public Reviews:**

**Reviewer #1 (Public review):**

We thank the reviewer for the positive evaluation of our paper and emphasizing its strengths in the Summary.

Weaknesses:(1) Could the authors provide a statement in the methods or results to clarify whether there were any changes in synaptic weight or other model parameters of the intact model to ensure locomotor activity in the hemisected model?

Such a statement has been inserted in Materials and Methods, section “Modeling”. Also, in the 1st paragraph of section “Spinal sensorimotor network architecture and operation after a lateral spinal hemisection”, we stated that no “additional changes or adjustments” were made.

(2) The authors should remind the reader what the main differences are between state-machine, flexor-driven, and classical half-center regimes (lines 77-79).

Short explanations/reminders have been inserted (see lines 80-83 of tracked changes document).

(3) There may be changes in the wiring of spinal locomotor networks after the hemisection. Yet, without applying any sort of plasticity, the model is able to replicate many of the experimental data. Based on what was experimentally replicated or not, what does the model tell us about possible sites of plasticity after hemisection?

Quantitative correspondence of changes in locomotor characteristics predicted by the model and those obtained experimentally provide additional validation of the model proposed in the preceding paper and used in this paper. This was our ultimate goal. None of the plastic changes during recovery were modeled because of a lack of precise information on these changes. The absence of possible plastic changes may explain the small discrepancies between our simulations and experimental data (see Supplemental Figures that have been added). However, the model only has a simplified description of spinal circuits without motoneurons and without real simulation of leg biomechanics. This limits our analysis or predictions of possible plastic changes within a reasonable degree of speculation. This issue is discussed in section: “Limitations and future directions” in the Discussion. We have also inserted a sentence: “The lack of possible plastic changes in spinal sensorimotor circuits of our model may explain the absence of exact/quantitative correspondences between simulated and experimental data.

(4) Why are the durations on the right hemisected (fast) side similar to results in the full spinal transected model (Rybak et al. 2024)? Is it because the left is in slow mode and so there is not much drive from the left side to the right side even though the latter is still receiving supraspinal drive, as opposed to in the full transection model? (lines 202-203).

This is correct. We have included this explanation in the text (lines 210-211 of tracked changes document).

(5) There is an error with probability (line 280).

This typo was corrected.

**Reviewer #2 (Public review):**
This is a nice article that presents interesting findings. One main concern is that I don't think the predictions from the simulation are overlaid on the animal data at any point - I understand the match is qualitative, which is fine, but even that is hard to judge without at least one figure overlaying some of the data.

We thank the Reviewer for the constructive comments. Figures showing the overlay of the experimental data with the modeling predictions have been included as figure supplements for Figures 5-7. This highlights how accurate the model predictions were.

Second is that it's not clear how the lateral coupling strengths of the model were trained/set, so it's hard to judge how important this hemi-split-belt paradigm is. The model's predictions match the data qualitatively, which is good; but does the comparison using the hemi-split-belt paradigm not offer any corrections to the model? The discussion points to modeling plasticity after SCI, which could be good, but does that mean the fit here is so good there's no point using the data to refine?

The model has not been trained or retrained, but was used as it was described in the preceding paper. Response: Quantitative correspondence of changes in locomotor characteristics predicted by the model and those obtained experimentally provide additional validation of the model proposed in the preceding paper and used in this paper. This was our ultimate goal. None of the plastic changes during recovery were modeled because of a lack of precise information on these changes. The absence of possible plastic changes may explain the small discrepancies between our simulations and experimental data (see figure supplements that have been added). However, the model only has a simplified description of spinal circuits without motoneurons and without real simulation of leg biomechanics. This limits our analysis or predictions of possible plastic changes within a reasonable degree of speculation. This issue is discussed in section: “Limitations and future directions” in the Discussion.

The manuscript is well-written and interesting. The putative neural circuit mechanisms that the model uncovers are great, if they can be tested in an animal somehow.

We agree and we are considering how we can do this in an animal model.

Page 2, lines 75-6: Perhaps it belongs in the other paper on the model, but it's surprising that in the section on how the model has been revised to have different regimes of operation as speed increases, there is no reference to a lot of past literature on this idea. Just one example would be Koditschek and Full, 1999 JEB Figure 3, where they talk about exactly this idea, or similarly Holmes et al., 2006 SIAM review Figure 7, but obviously many more have put this forward over the years (Daley and Beiwener, etc). It's neat in this model to have it tied down to a detailed neural model that can be compared with the vast cat literature, but the concept of this has been talked about for at least 25+ years. Maybe a review that discusses it should be cited?

We have revised the Introduction to include the suggested references.

Page 2, line 88: While it makes sense to think of the sides as supraspinal vs afferent driven, respectively, what is the added insight from having them coupled laterally in this hemisection model? What does that buy you beyond complete transection (both sides no supra) compared with intact?

We are trying to make one model that could reproduce multiple experimental data in quadrupedal locomotion, including genetic manipulations with (silencing/removal) particular neuron types (and commissural interneurons), as pointed out in the section “Model Description” in the Results. These lateral connections are critical for reproducing and explaining other locomotor behaviors demonstrated experimentally. However, even in this study, these lateral interactions are necessary to maintain left-right coordination and equal left-right frequency (step period) during split-belt locomotion and after hemisection.

I can see how being able to vary cycle frequencies separately of the two limbs is a good "knob" to vary when perturbing the system in order to refine the model. But there isn't a ton of context explaining how the hemi-section with split belt paradigm is important for refining the model, and therefore the science. Is it somehow importantly related to the new "regimes" of operation versus speed idea for the model?

We did not refine the model in this paper. We just used it for new simulations. The predictions strengthen the organization and operation of the model we recently proposed.

Page 5, line 212: For the predictions from the model, a lot depends on how strong the lateral coupling of the model is, which, in turn, depends on the data the model was trained on. Were the model parameters (especially for lateral coupling of the limbs) trained on data in a context where limbs were pushed out of phase and neuronal connectivity was likely required to bring the limbs back into the same phase relationship? Because if the model had no need for lateral coupling, then it's not so surprising that the hemisected limbs behave like separate limbs, one with surpaspinal intact and one without.

Please see our response above concerning the need for lateral interactions incorporated to the model.

Page 8, line 360: The discussion of the mechanisms (increased influence of afferents, etc) that the model reveals could be causing the changes is exciting, though I'm not sure if there is an animal model where it can be tested in vivo in a moving animal.

We agree it may be difficult to test right now but we are considering experimental approaches.

Page 9, line 395: There are some interesting conclusions that rely on the hemi-split-belt paradigm here.

We agree with this comment. Thanks.

**Reviewer #2 (Recommendations for the authors):**
Figures: Why aren't there any figures with the simulation results overlaid on the animal data?

We followed this suggestion. Figures showing the overlay of the experimental data with the modeling predictions have been included as figure supplements.